# Support for the Vulnerability Management Process Using Conversion CVSS Base Score 2.0 to 3.x

**DOI:** 10.3390/s23041802

**Published:** 2023-02-06

**Authors:** Maciej Roman Nowak, Michał Walkowski, Sławomir Sujecki

**Affiliations:** Department of Telecommunications and Teleinformatics, Wroclaw University of Science and Technology, 50-370 Wroclaw, Poland

**Keywords:** CVSS standard, machine learning, security of IT systems

## Abstract

COVID-19 forced a number of changes in many areas of life, which resulted in an increase in human activity in cyberspace. Furthermore, the number of cyberattacks has increased. In such circumstances, detection, accurate prioritisation, and timely removal of critical vulnerabilities is of key importance for ensuring the security of various organisations. One of the most-commonly used vulnerability assessment standards is the Common Vulnerability Scoring System (CVSS), which allows for assessing the degree of vulnerability criticality on a scale from 0 to 10. Unfortunately, not all detected vulnerabilities have defined CVSS base scores, or if they do, they are not always expressed using the latest standard (CVSS 3.x). In this work, we propose using machine learning algorithms to convert the CVSS vector from Version 2.0 to 3.x. We discuss in detail the individual steps of the conversion procedure, starting from data acquisition using vulnerability databases and Natural Language Processing (NLP) algorithms, to the vector mapping process based on the optimisation of ML algorithm parameters, and finally, the application of machine learning to calculate the CVSS 3.x vector components. The calculated example results showed the effectiveness of the proposed method for the conversion of the CVSS 2.0 vector to the CVSS 3.x standard.

## 1. Introduction

COVID-19 forced a number of changes in many areas of people’s lives, which directly contributed to the emergence of the cyberpandemic phenomenon [1]. From the point of view of organisations, both private and public, the complete or even partial transfer of professional activity to cyberspace resulted in a larger exposure to cyberattacks. As the existing information and communication technology (ICT) infrastructure had to handle increased network traffic, it became more susceptible to various types of attacks, e.g., Distributed Denial of Service (DDoS) [2] attacks. Remote work often forced organisations to reduce the level of security for access to data, because employees used access to the organisation’s infrastructure from their own devices, based on the Bring Your Own Device (BYOD) policy [3]. Not all organisations were prepared for this and had Mobile Device Management (MDM)/Enterprise Mobile Management (EMM) or Unified Endpoint Management (UEM) systems implemented. Additionally, malware authors have stepped up cryptojacking and ransomware development activities. In 2021, the number of new cryptojacking programs increased by 75% over the year, while ransomware programs increased by 42% [4]. In this context, a particularly disturbing fact is that, in 2021, a randomly selected company needed on average 280 days to detect and respond to a cyberattack, whereas between 2019 and 2021, the average cost of a data breach increased from USD 3.92 million to USD 4.24 million [5,6] and continues to increase [7]. New vulnerabilities add to an already large number of existing ones [4], which further complicates the process of the detection and timely removal of critical vulnerabilities needed to ensure the security of ICT infrastructure.

Vulnerability management and prioritisation are therefore essential elements in the daily activities of every reputable company. Specifically, the topic of prioritising threats has long been discussed in the scientific literature related to cybersecurity [8,9,10]. One of the weak points of currently implemented approaches is that each organisation handles the problem differently, i.e., in accordance with its internal security policy [11]. Further, commercial vulnerability management systems are very expensive and do not inform users about the details of the vulnerability prioritisation procedure. Additionally, in order to assess the security level of the ICT infrastructure, organisations use metrics proposed in 2010 by the Center for Internet Security (CIS) [12], while in order to meet the CIS requirements, the most-frequently chosen standard is based on the Common Vulnerability Scoring System (CVSS) 2.0 base score (BS). However, a new standard, i.e., the CVSS 3.x (BS), is already available and provides a better assessment of vulnerability criticality.

The Common Vulnerability Scoring System (CVSS) is a standard that describes the vulnerability characteristics of software [13] and is used to assess the severity of a threat on a scale from 0 to 10. The CVSS 2.0 standard was introduced in 2007 [14], while the subsequent version—CVSS 3.0 [15]—was introduced in 2015. In 2019, CVSS 3.0, due to significant problems caused by the possibility of different interpretations [16], was replaced by the CVSS 3.1 [17] version. Although CVSS 3.1 is the most-recent version of the standard, it has not yet been introduced widely. This is because, due to a large total number of all vulnerabilities, not all CVSS scores could have been recalculated in accordance with the guidelines of the latest 3.1 standard, even though the new CVSS standard offers better assessment of vulnerability criticality [18].

Regardless of the version, the vulnerability rating is influenced by three scoring categories—Basic (BS), Time (TS), and Environmental (ES). The latter two categories are optional and are used in the case of a very thorough analysis. In our considerations, we focused on the BS for two versions, 2.0 and 3.x. The most-important differences between the standards are presented in Table 1. Despite the changes in the nomenclature of both standards, the parameters recorded in the individual rows of Table 1 are functionally consistent. The CVSS 2.0 base vector is described by 6 components, which can take one of the three values (the notations of metric values are written next to each parameter after a slash), while in CVSS 3.x, one needs to know as many as 8 parameters, i.e., two additional parameters: S and UI. Another difference is the number of metric values that can be assumed by the parameters AV and AC of the CVSS 3.x base vector, i.e., 4 and 2, respectively.

There is also a discrepancy between the Qualitative Severity Rating Scale (QSRS) presented in Table 2.

Despite some ambiguities, both CVSS 2.0 and CVSS 3.x are used simultaneously by IT operators [11]. However, the lack of complete information on the BS 3.x scores for all detected threats affects the operation of the currently used vulnerability prioritisation systems. Namely, their correct operation relies mainly on the CVSS 2.0 standard and vulnerability scanners, which calculate only the BS according to the CVSS 2.0 standard. Thus, important information related to infrastructure details and organisation context is omitted [19,20,21]. Further, it is known that the CVSS 2.0 ES does not provide an accurate measure of IT infrastructure security, because the Target Distribution (TD) parameter underestimates the assessment of all detected vulnerabilities. The CVSS 3.x standard, on the other hand, gives a more accurate assessment of threats’ criticality including organisation context, which directly translates into a higher level of protection against hacker attacks [18,22]. Summarising, there are clear benefits to using the newer 3.x standard. However, the NIST statistical data [23] indicate that, on 26 January 2023, a total of 205,539 vulnerabilities were recorded in the database, of which 180,086 have a CVSS 2.0 BS rating, while only 120,114 have a CVSS 3.x BS rating. This means that 25,453 vulnerabilities have not been assessed at all, while 59,972 require conversion to CVSS 3.x, which is about 33% of all known vulnerabilities. It is hard to say exactly how long NIST will take to recalculate old vulnerabilities if more than 12% of new vulnerabilities have not yet been assessed. Thus software tools are needed for the fast and efficient calculation of CVSS scores for all vulnerabilities.

In the literature, one can find several threat prediction methods using machine learning [24,25,26] to help organisations assess the criticality of vulnerabilities. Additionally, several algorithms have been developed for the automatic estimation of the CVSS BS and the vulnerability description in order to speed up the process of CVSS base score estimation and to reduce the time needed for score publication in National Vulnerability Databases (NVDs) [27,28]. Some authors have also considered the temporal [29] and environmental [30,31] categories. Here, we used machine learning algorithms to convert the base score from the CVSS 2.0 to the CVSS 3.x standard by estimating not only the final score, as in the mentioned prior work, but all components of the CVSS 3.x vector, which is the main unique contribution of this study.

## 2. Research Concept

In this section, we describe the method used in this contribution to convert the BS CVSS 2.0 to the 3.x standard. The proposed procedure includes four steps—data acquisition, the selection of the training set, the selection of the Machine Learning (ML) algorithms, and their testing [16,17].

The three last steps are critical to correct mapping of vectors from the CVSS 2.0 to the CVSS 3.x standard and, hence, present an integral entity, which we refer to in the rest of the text as “Mapping using ML”. The final BS CVSS 3.x value is obtained by substituting the eight components of the CVSS 3.x vector, determined via the ML-facilitated conversion from the CVSS 2.0 standard into the formulae given in the CVSS 3.x specification

Figure 1 illustrates the process of mapping the extended CVSS 2.0 vector onto the classes of the AV parameter of the CVSS 3.x vector. It is noted that every step of the mapping procedure is linked to the relevant section containing its detailed description. Next, we discuss in detail the subsequent steps of the mapping procedure.

### 2.1. Data Acquisition

The first step in the data acquisition is to obtain the list of vulnerabilities using the open-source Vulnerability Management Center (VMC) [32,33] software from the National Institute of Standards and Technology (NIST) database (September 2021). The database provides information on all known 73,179 vulnerabilities with both BS CVSS 2.0 and 3.x ratings. This represents approximately 49% of all vulnerabilities available from the NIST database. The remaining 51% of the vulnerabilities do not have the CVSS 3.x rating. Then, using a script implemented in the Python programming language, 2.0 and 3.x BS vectors were created containing the values of the BS vector components according to the specification given by Table 1 and the assigned BS score value calculated according to the CVSS specification [16,17]. Thus, we obtained seven components of the original CVSS BS 2.0 vector, which consisted of six base score components and one component containing the CVSS 2.0 base score (Figure 1). Preliminary studies on the CVSS 2.0 to 3.x conversion using ML algorithms showed that using a BS vector containing only the BS components as specified by Table 1 gives a very low conversion accuracy. Therefore, the CVSS 2.0 BS vector was complemented additionally with either 50 or 100 elements. The additional elements of the extended CVSS 2.0 vector were obtained from vulnerability descriptions by counting the number of occurrences for selected keywords (from the 3,458,874 base words obtained). The necessary text processing was performed using the Natural Language Toolkit (NLTK) Phyton library [34]. All stop words were removed, and stemming was used to normalise the text. We used the Porter stemmer [35] based on a set of heuristics that are used to remove common suffixes from words. The additional elements were obtained from vulnerability descriptions by counting the number of occurrence for selected keywords. The necessary text processing was performed using the Natural Language Toolkit (NLTK) Phyton library. The keywords are listed in the order corresponding to the number of occurrences of a particular word. First, 50 words were selected to form the 50-component NLTK vector, while the 100-component NLTK vector was obtained from the 50-component one by adding further (less-often-occurring) words from the list. Therefore, the 100-component NLTK vector is an extension of the 50-component one. The CVSS BS 2.0 extended vector was formed by concatenating the original CVSS BS 2.0 vector with an NLTK vector (Figure 1). This procedure yielded two sets, which were used for creating the learning sets. Both sets contained 73,179 vectors. However, one of them consisted of 57 component vectors, while the over one of 107 component vectors. The list of the selected keywords, input data, and Phyton scripts used for the creation of both sets of vectors have been placed in the public repository [36] for the interested reader.

### 2.2. Procedure for Creating the Training Set

The data obtained through the data acquisition as described in the preceding Section were unbalanced (Table 3). Namely, for each class corresponding to a particular value of the CVSS 3.x BS vector component, one has a widely varying number of CVSS 2.0 BS vectors that map onto it. For instance, to the CVSS 3.x BS vector AV component class N map 841 CVSS 2.0 BS vectors, while to class P 53732 CVSS 2.0 BS vectors [37].

Therefore, a proprietary undersampling method as described in [37] was used to obtain a more balanced training set. Using the undersampling method, a common training set was created from the extended CVSS 2.0 vectors, assuming that, for each class of the CVSS 3.x BS vector component, we selected only 80 corresponding CVSS 2.0 BS vectors. In order to achieve this, we first selected the reference CVSS 2.0 extended vector. For this purpose, we took all vectors that corresponded to a specific class of the CVSS 3.x vector and list them in a table. Thus, the rows of the table are the CVSS 2.0 extended vectors, and the columns give the value of the specific extended CVSS 2.0 vector component. For instance, the first column gives the values of the AV component of CVSS 2.0 extended vector (Table 1). In the next step for each column, we selected the component value that occurred most often, i.e., the median value. This procedure once completed for all CVSS 2.0 extended vector components yielded a CVSS 2.0 extended vector, which was assumed to be the reference vector. Then, 80 CVSS 2.0 extended vectors were selected, which were “nearest” to the reference vector in the sense of the correlation function. In this particular implementation, the Matlab correlation function “corrcoef” was used and the vectors were selected, for which the value of the correlation function was nearest to 1. Then, we repeated the procedure for all classes of the CVSS 3.x BS vector components, each time creating a set consisting of 80 elements. In the final step, we combined all the thus-created sets into a training set, whereby all duplicates were removed. This procedure was performed two times, once using 57 component CVSS 2.0 extended vectors and once using 107 component CVSS 2.0 extended vectors. This created two learning sets. The set created using the extended CVSS 2.0 vector with a vector length of 57, after removing the duplicates, had 1736 elements, while the one assembled with 107 element vectors consisted of 1737 components. Both sets were used for the initial study of the machine learning algorithm accuracy for each CVSS 3.x vector class. Then, for each CVSS 3.x class, the set that provided better performance was selected and was used consistently in the simulations described in Section 3. The selection procedure is describe in more detail in Section 2.4 and in [37]. Table 4 provides the information on the selection of the specific CVSS 2.0 extended vector set for each component of the CVSS 3.x vector.

### 2.3. Machine Learning Algorithms

Five machine learning algorithms were selected for the research—k-Nearest Neighbours (kNN) with the Euclidean metric (E) and Cosine similarity (C), the Naive Bayes classifier (NB), the Probabilistic Neural Network (PNN), Soft Independent Modelling of Class Analogy (SIMCA), and Support Vector Machine with a Kernel function (KSVM) in two configurations—with a Linear kernel function (LIN) and a Gaussian Radial Basis Function (GRBF). Except for the last mentioned algorithm, the remaining ones are well known and well described in the available literature [38].

The KSVM GRBF uses a two-fold cross-validation, the Misclassification Rate (MCR), and the Nelder–Mead [39] algorithm with a set of 50 starting points and kNN with a cosine (C) metric for the final classification of unrecognised samples.

In short, the main criterion used for testing and validating the model is the Misclassification Rate (MCR). The curvature of the nonlinear separating hypersurface can be controlled by the kernel function and its related parameters. The kernel function takes the form of the Gaussian Radial Basis Function (GRBF): $k\left(x_{1},x_{2}\right)=\exp\left\{\frac{\left|\right|x_{1}-x_{2}{\left|\right|}^{2}}{2\sigma^{2}}\right\}$, where $\sigma^{2}>0$. Let $C_{\xi }$ be the soft-margin parameter used to separate outlier samples. Let $p=\{\sigma^{2},C_{\xi }\}$ be a vector of the parameters to be estimated. To estimate the vector *p*, the two-fold CV was applied separately to each class. Thus, the problem of the data-driven estimation of the parameters comes down to MCR minimalisation. To finally optimise *p*, we chose the Nelder–Mead (NM) method [39]. If a testing sample cannot be identified by any trained classifier, we can apply kNN with a Cosine (C) metric (three nearest neighbours) for the final classification classifier. In the experiments, we used the Sequential Minimal Optimisation (SMO) algorithm to solve the QP problem associated with finding the best separating hyperplane. Additionally, the One-vs.-One (OvO) approach was used for parameters with at least three classes. A more detailed description of how the algorithm works is included in the publication [40].

Compared to our previous publication [37], we extended the scope of the basic research with two ML algorithms—SIMCA and PNN. The use of a wide range of ML algorithms allows one to match the most-appropriate method to the specific problem. Based on Table 1, it can be seen that, for some parameters of the CVSS 3.x vector (AC, S, UI), we are dealing with a binary classification problem, where we expect that less computationally complex algorithms can be used. A more difficult decision problem arises when there are more classes. This situation is the case for the remaining parameters of the CVSS 3.x vector, i.e., AV, PR, C, I, and A. For these parameters, the attempt to separate the classes becomes much more complicated, because one cannot assume that the classes are linearly separable. This necessitates using algorithms that allow for tuning a set of parameters responsible for the correct classification, e.g., KSVM with a kernel function. In the process of ML method selection, training speed is also considered, because it directly translates into greater efficiency in real software implementations. If similar results of the mapping of the extended BS CVSS 2.0 vector to a specific component of the BS CVSS 3.x vector are achieved using a fast parametric algorithm, then the use of a slower non-parametric algorithm is considered not justified.

### 2.4. Algorithm Selection Procedure—Testing

The parameters of all machine learning algorithms were optimised for the classification of the individual components of the CVSS 3.x BS vector. The different lengths of the training vectors and the number of components for preprocessing by Principal Component Analysis (PCA) were taken into account. The preselection procedure involved determining the accuracy of the algorithm classification while testing with a set of 100 randomly selected vectors (independent of the training set described in Section 2.2). The task was repeated 10 times, and up to three algorithms were qualified for the further part of the research. In the next step, the quality of the models was assessed by carrying out a test involving a five-fold cross-validation of the training set described in Section 2.2. Based on the averaged MCR values, sensitivity, precision, and F1-score, the best classification models were matched. The confirmation of the accuracy and repeatability of the conversion of CVSS 2.0 vectors into classes of CVSS 3.x base metrics parameters was performed using a three-stage testing for sets of various sizes, i.e., consisting of 1000, 10,000, and 71,000 elements, whilst repeating the task 50, 50, and 10 times, respectively.

Table 4 summarises the final assignment of the machine learning algorithms to the classification of individual components of the CVSS 3.x base vector. In addition, information is given on the lengths of the training vectors and the number of PC components (PCs) with which the accuracy and F1-score maximisation was obtained.

**Table 4 sensors-23-01802-t004:** Assignment of machine learning algorithms to the classification of the components of the CVSS 3.x base vector with the number of PCs and the length of the training vectors.

CVSS 3.x BS Vector Component	Algorithm	Number of PCs	Vector Length
AV	KSVM GRBF	35	107
AC	PNN	25	107
PR	KSVM GRBF	24	57
S	NB	38	107
UI	PNN	41	107
C	KSVM GRBF	43	57
I	KSVM GRBF	32	107
A	PNN	48	107

### 2.5. CVSS 3.x BS Calculation

The components of the CVSS 3.x vector obtained using machine learning mapping of an extended CVSS 2.0 vector formed the basis for the calculation of the BS. The numbers representing the class indices were converted to metric values following the procedure described in the Section 7.4 CVSS documentation [17]. The base score CVSS 3.x formula depends on the sub-formulas for the Impact Sub-Score (*ISS*), impact, and exploitability, as defined below by Formulae (Equation 1)–(Equation 4) given below. *ISS* is given by:(1)$$\begin{array}{c} ISS=1-[(1-C)\cdot (1-I)\cdot (1-A)] \end{array}$$

The key parameter for the calculation of the CVSS 3.x BS is the Scope (S), which determines the use of one of the two methods of calculating the impact, described by Formulae (Equation 2) and (Equation 3).

If the scope is unchanged, then
(2)$$\begin{array}{c} Impact=6.42\cdot ISS \end{array}$$

If the scope is changed, then
(3)$$\begin{array}{c} Impact=7.52\cdot \left(ISS-0.029\right)-3.25\cdot {\left(ISS-0.02\right)}^{15} \end{array}$$

The next parameter that needs to be calculated is the *exploitability*:(4)$$\begin{array}{c} Exploitability=8.22\cdot AV\cdot AC\cdot PR\cdot UI \end{array}$$

The final formula for the calculation of the CVSS 3.x base score can take two forms, either (Equation 5) or (Equation 6), depending on the value of the Scope (S) parameter.

If the scope is unchanged:(5)$$\begin{array}{c} BS=Roundup(Minimum[(Impact+Exploitability),10]) \end{array}$$

If the scope is changed:(6)$$\begin{array}{c} BS=Roundup(Minimum[1.08\cdot (Impact+Exploitability),10]) \end{array}$$
where Minimum returns the smaller of its two arguments. Roundup returns the smallest number, specified to 1 decimal place, that is equal to or higher than its input. A special case occurs when Impact<=0, then BS = 0.

## 3. Results

As described in Section 2, the training set consisted either of 1736 or 1737 elements. Those elements were removed from the set of all 73,179 CVSS 2.0 vectors. The remaining elements were used to test the ML model and form a very large testing set, whereby three cases were considered. Namely, from the entire testing set, we selected smaller testing sets consisting of 1000-, 10,000-, and 71,000-element CVSS 2.0 vectors. For instance, in the case of the 1000-element set, we randomly selected using a uniformly distributed pseudorandom number generator 1000 elements from the original testing set and calculated all the performance metrics, i.e., precision, recall, F1-score, and accuracy. The general aim was to maximise the precision and recall whilst keeping the accuracy as high as possible. Similarly, we selected 10,000 and 71,000 elements. For each size of the testing set, we repeated this procedure several times and obtained the final result in the form of the average value and standard deviation. The number of repetitions was, respectively, 50, 50, and 10 for the sets consisting of 1000, 10,000, and 71,000 elements. Following this procedure, the average accuracy of the classification and its standard deviation were calculated for each component of the CVSS 3.x BS vector (Table 5). We note that the ML algorithms used for each component of the CVSS 3.x BS vector are listed in Table 4.

The results presented in Table 5 show that, within the same component of the CVSS 3.x vector, the mean values differed from each other by less than 0.75%. Further, for smaller testing sets, we generally observed larger values of the standard deviation, with the exception of the components AV and C, whereby some anomalies were present. Furthermore, we calculated the median values for the accuracy and present these results in Figure 2 (the red line gives the median value). Figure 2 shows the calculated statistics of classifying the CVSS 3.x vector parameters for those selected in the Table 4 6 algorithms. For these calculations, the testing sets consisted of 71,000 elements, while the tests were repeated 10 times. It can be noticed from the results shown in Figure 2 that, regardless of the parameter and the size of the testing set, the median accuracy of the classification was always greater than 90%.

We also note that the results presented in Table 5 have small values of the standard deviation with the largest values of the standard deviation observed for the PR and AV components of the CVSS 3.x BS vector. The results from Figure 2 show also that the PR component has two outliers, which occurred for more than three percent below the median. Outliers occurred also for components S, C, and A. However, in those cases, they were less critical. Comparing the results shown in [40], there was a noticeable improvement in the observed accuracy for all the components of the CVSS 3.x BS vector. In particular, the lower and upper quartiles for the components S and A significantly reduced. This improvement is critical for the accurate calculation of the CVSS 3.x base score, as will be explained in more detail in the next paragraph.

The CVSS 3.x BS is obtained by substituting the calculated values of the BS CVSS 3.x vector components into the mathematical Formulae (Equation 5) and (Equation 6). The thus-obtained CVSS 3.x BS has a lower conversion accuracy, i.e., 62.4–62.8%, when compared with at least a 90% accuracy achieved for each CVSS 3.x BS vector component, treated separately. One can observe that the accuracy for the calculated CVSS 3.x BS value is approximately equal to the product of the classification accuracy obtained for each component, which one should expect. The key parameter for the calculation of the CVSS 3.x BS is the Scope (S), which determines the use of one of the two methods of calculating the BS (Section 2.5).

The quality of the classification was estimated by calculating the precision, recall, and F1-score. The results obtained from the 10 testing trials of the algorithms with a randomly selected 71,000-element testing set are shown in Table 6. Generally, the quality of the prediction was good. However, the presented statistics indicate that some imperfections were still present despite all the effort invested in the tuning of the model hyperparameters. Especially, there were problems with the parameter A for class index = 2, the parameter AV for class index = 1, and the PR for class index = 3. In order to address these difficulties, some further development of the proposed method will be necessary in the future.

In order to explain why it is difficult to achieve high accuracy for the CVSS 3.x BS, we plot in Figure 3 a histogram, which shows the number of class index combinations (number of scenarios) for the CVSS 3.x base vector that map onto a particular value of the base score. We note that the total number of all class index combinations was 2592, while the value of BS was in the range of 0–10 with a resolution of 0.1, i.e., there were theoretically 101 distinct values. The results from Figure 3 shows that the number of class combinations corresponding to the individual values of the CVSS 3.x BS score vary with the score value, i.e., the distribution is not uniform (Figure 3). These results show also that, for some CVSS 3.x BS values, there are no corresponding CVSS 3.x vectors. For instance, there is a large gap between the BS of 0.1 and 1.5. In other words, no matter how one selects the values of the CVSS 3.x BS vector component flags, the calculated score will not fall between 0.1 and 1.5. Furthermore, for BS = 9.5 and BS = 9.7, there is no corresponding combination. Further, for BS = 9.8, we have only one combination. In contrast, for BS = 5.7, there are as many as 79 different base vectors, while the value of BS = 0 can be obtained using 96 different flag combinations. The particularly large number of CVSS 3.x BS vectors mapping onto the 0 score follows from the fact that a zero BS score was obtained when C = I = A = 0.

Figure 4 shows the confusion matrix of the vulnerability categories for the conversion of the CVSS 2.0 vector to the CVSS 3.x standard obtained for 71,000 vulnerabilities. The conversion was performed 10 times; therefore, the matrix shows the cumulative values (710,000 observations), i.e., the results for all 10 repetitions were collected together to calculate the numbers shown in Figure 4 for each score. It can be seen that our solution worked best with the classification of the CVSS 3.x base score for the critical category, whereby 87,248 vulnerabilities were classified correctly as having critical values of the BS. However, 11,743 critical vulnerabilities were classified incorrectly, i.e., as having a lower BS. This corresponds to the values of precision and recall equal to 91% and 88.1%, respectively. It is important to note that a very good performance for the critical category was achieved despite of the fact that most vulnerabilities belonged to the high and low categories. The worst performance was achieved for the lowest categories. However, in these cases, the results of the misclassification resulted in the overestimation of the score, which has much less severe consequences than underestimation. Furthermore, as there were no vulnerabilities in the data base, which would have BS = 0, the first row of the confusion matrix is empty. However, the elements of the first column are not equal to zero since some vulnerabilities were classified by ML algorithms as having BS = 0. It is noted, though, that the number of such vulnerabilities is relatively low. For instance, for critical vulnerabilities, only two were predicted to have a 0 score. Nonetheless, we note that even one critical vulnerability classified as having BS = 0 may have severe consequences for the security of the system.

In order to mitigate the consequences of the false categorisation of critical vulnerabilities as a one with a zero score, one can introduce for all vulnerabilities mapped onto a zero score an automatic check, which would detect the presence of a 0 0 0 sequence for the scores of the C I A (Impact = 0) components of the CVSS 3.x vector. From the description of the formula for the calculation of the CVSS 3.x score, one can deduce that such a combination necessarily maps the BS score onto the zero value. In such a case, one can raise one of the C I A component scores one category up. For instance, change C = 0 to C = 1. Such a modification only slightly changes the confusion matrix for all higher scores, but eliminates effectively predicted zero scores. Figure 5 shows the confusion matrix obtained after introducing such an extension of the original vulnerability classification procedure. We note that, due to a symmetry present within the formula for the calculation of CVSS 3.x BS, it does not matter which component score we raise by 1 from the three components considered, i.e., C, I or A. This procedure works with equal effect for all three cases.

## 4. Conclusions

The presented research results indicated that Machine Learning (ML) algorithms can be used to aid the process of classification for the metric parameters of the CVSS 3.x base vector. For the ML algorithms developed in this contribution, the median of the classification accuracy in the worst case, i.e., for the parameter AV, is 90%. To the best of our knowledge, these are the best results obtained by ML algorithms for the CVSS 3.x BS classification published in the available literature so far. The selected ML algorithms for the observations belonging to the two lowest categories (informative and low) tended to overestimate the results, which helped in reducing the consequences of underestimating the CVSS 3.x BS. It is also noted that the selected ML algorithms predicted the vulnerability categories with the highest precision for the critical category (91% precision). The total conversion precision for the entire CVSS 3.x score was (62–63)%. A notable reduction of the final CVSS 3.x BS accuracy when compared with the individual components resulted from the fact that the final value of the CVSS 3.x BS was calculated using mathematical formulas linking the eight vector components. Finally, we note that the described ML-based solution can be applied to replace the CVSS 2.0 standard with the CVSS 3.x one during the vulnerability management process in vulnerability management centres [22]. This is important because, using the newer CVSS 3.x standard, it is possible to more accurately estimate the level of asset security, especially when taking into account the context of a particular organisation environment. Finally, despite the large improvement of the conversion accuracy, confirmed by the results presented in Section 3, further refinement of the machine learning algorithms aimed at eliminating the shortcomings pertaining to the parameters AV, PR, and A will be the subject of our future work.

## Figures and Tables

**Figure 1 sensors-23-01802-f001:**
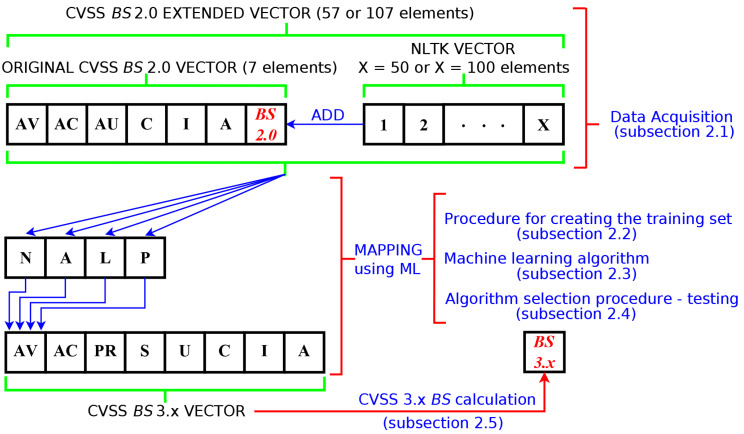
Schematic diagram showing the mapping of the CVSS 2.0 extended vectors onto classes of the AV component of the CVSS 3.x BS vector.

**Figure 2 sensors-23-01802-f002:**
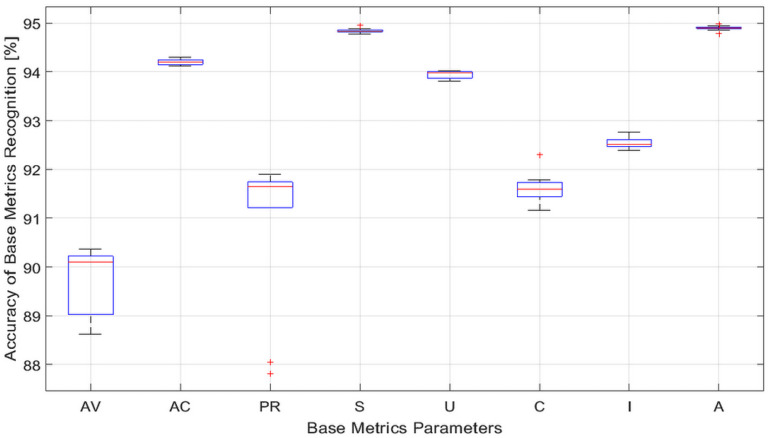
The classification statistics of CVSS 3.x vector parameters using the parameters from Table 4.

**Figure 3 sensors-23-01802-f003:**
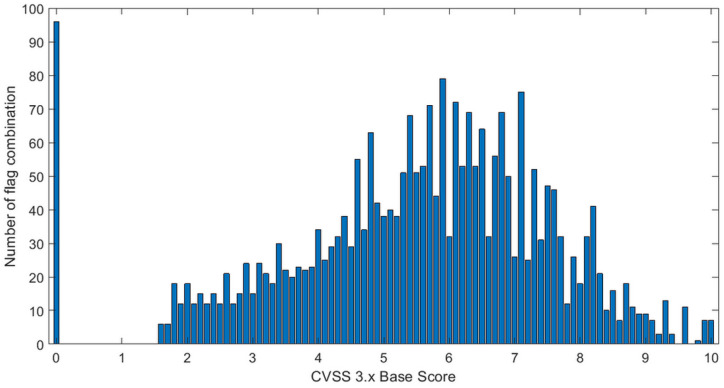
Number of combinations (scenarios) corresponding to the baseline scores.

**Figure 4 sensors-23-01802-f004:**
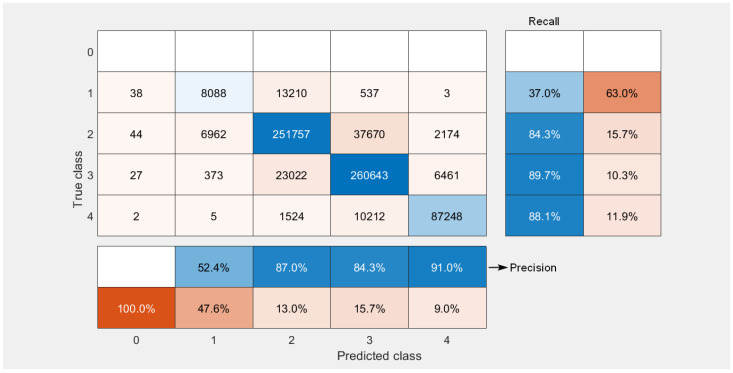
Confusion matrix for qualitative severity rating scale classification.

**Figure 5 sensors-23-01802-f005:**
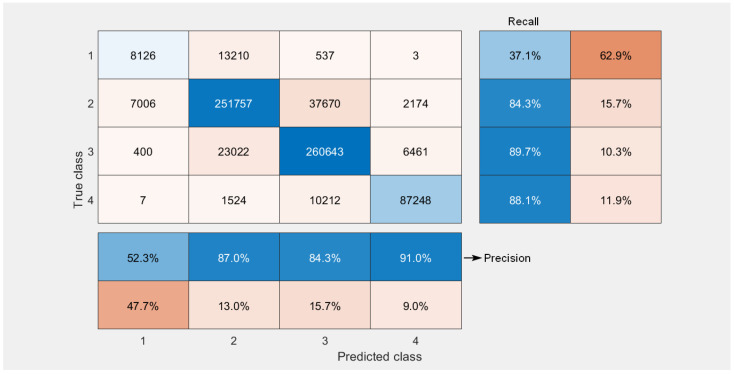
Confusion matrix for qualitative severity rating scale classification after modification.

**Table 1 sensors-23-01802-t001:** Differences in the construction of the CVSS 2.0 and CVSS 3.x base vectors.

CVSS 2.0 Standard	CVSS 3.x Standard
Access Vector (AV)/L, A, N	Attack Vector (AV)/N, A, L, P
Access Complexity (AC)/H, M, L	Attack Complexity (AC)/L, H
Authentication (Au)/M, S, N	Privileges Required (PRs)/N, L, H
	User Interaction (UI)/N, R
	Scope (S)/U, C
Confidentiality Impact (C)/N, P, C	Confidentiality Impact (C)/H, L, N
Integrity Impact (I)/N, P, C	Integrity Impact (I)/H, L, N
Availability Impact (A)/N, P, C	Availability Impact (A)/H, L, N

**Table 2 sensors-23-01802-t002:** Differences in vulnerability categories for CVSS 2.0 and CVSS 3.x.

Rating	CVSS 2.0 Standard	CVSS 3.x Standard
0.0		None
0.1–3.9	Low	Low
4.0–6.9	Medium	Medium
7.0–8.9	High	High
9.0–10.0		Critical

**Table 3 sensors-23-01802-t003:** The number of CVSS 2.0 vectors that map onto a specific class of the CVSS 3.x vector.

CVSS 3.x *BS* Vector Component	Class Index	No of CVSS 2.0 Vectors
AV	1	841
	2	1573
	3	17,033
	4	53,732
AC	1	6021
	2	67,158
PR	1	4452
	2	19,380
	3	49,347
S	1	12,156
	2	61,023
UI	1	26,449
	2	46,730
C	1	14,074
	2	15,932
	3	43,173
I	1	12,818
	2	22,831
	3	37,530
A	1	1883
	2	28,440
	3	42,856

**Table 5 sensors-23-01802-t005:** The average accuracy of the classification of the CVSS 3.x base vector parameters achieved for the testing set of 1000, 10,000, and 71,000 elements and testing multiplicities of 50, 50, and 10 times, respectively.

Base Metrics CVSS 3.x	Average Accuracy Determined for the Testing Set Of		
	1000 Vectors	10,000 Vectors	71,000 Vectors
AV	90.11 ± 1.17	90.11 ± 0.42	89.73 ± 0.66
AC	94.40 ± 0.75	94.22 ± 0.19	94.21 ± 0.06
PR	90.53 ± 1.74	90.39 ± 1.60	90.89 ± 1.49
S	94.19 ± 0.75	94.18 ± 0.24	94.84 ± 0.05
UI	93.90 ± 0.81	94.00 ± 0.23	93.95 ± 0.07
C	90.93 ± 0.87	90.88 ± 0.99	91.62 ± 0.29
I	92.55 ± 0.88	92.65 ± 0.25	92.53 ± 0.11
A	94.74 ± 0.75	94.87 ± 0.22	94.90 ± 0.05

**Table 6 sensors-23-01802-t006:** Precision, recall, F1-score, and MCR calculated for the algorithms listed in Table 4 obtained from the 10 testing trials of the algorithms with a randomly selected 71,000-element testing set.

CVSS 3.x BS Vector Component	Class Index	Precision	Recall	F1 Score
AV	1	0.1666 ± 0.0843	0.2200 ± 0.0385	0.1763 ± 0.0345
	2	0.7899 ± 0.0648	0.7972 ± 0.0368	0.7905 ± 0.0223
	3	0.7656 ± 0.0632	0.9707 ± 0.0234	0.8544 ± 0.0358
	4	0.9501 ± 0.0142	0.9416 ± 0.0112	0.9457 ± 0.0015
AC	1	0.9872 ± 0.0004	0.9510 ± 0.0007	0.9688 ± 0.0004
	2	0.4274 ± 0.0071	0.7474 ± 0.0049	0.5438 ± 0.0058
PR	1	0.9729 ± 0.0057	0.9797 ± 0.0169	0.9762 ± 0.0059
	2	0.8917 ± 0.0904	0.7991 ± 0.0149	0.8400 ± 0.0366
	3	0.2602 ± 0.0823	0.5216 ± 0.1036	0.3289 ± 0.0350
S	1	0.9865 ± 0.0005	0.9536 ± 0.0012	0.9697 ± 0.0006
	2	0.7544 ± 0.0050	0.9159 ± 0.0025	0.8273 ± 0.0030
UI	1	0.9260 ± 0.0010	0.9778 ± 0.0010	0.9512 ± 0.0009
	2	0.9627 ± 0.0013	0.8800 ± 0.0010	0.9195 ± 0.0009
C	1	0.9690 ± 0.0058	0.9850 ± 0.0012	0.9769 ± 0.0026
	2	0.9895 ± 0.0058	0.8420 ± 0.0130	0.7754 ± 0.0049
	3	0.9589 ± 0.0046	0.9108 ± 0.0027	0.9343 ± 0.0019
I	1	0.9820 ± 0.0017	0.9514 ± 0.0013	0.9665 ± 0.0011
	2	0.8316 ± 0.0043	0.8559 ± 0.0047	0.8435 ± 0.0011
	3	0.9248 ± 0.0012	0.9340 ± 0.0020	0.9294 ± 0.0008
A	1	0.9783 ± 0.0006	0.9541 ± 0.0011	0.9660 ± 0.0007
	2	0.0545 ± 0.0049	0.3229 ± 0.0285	0.0933 ± 0.0081
	3	0.9667 ± 0.0007	0.9497 ± 0.0008	0.9581 ± 0.0005

## Data Availability

Not applicable.

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
