# Peer review of "Support for the Vulnerability Management Process Using Conversion CVSS Base Score 2.0 to 3.x"

_sensors, 2023, doi:10.3390/s23041802_

Round 1

Reviewer 1 Report

The authors propose a machine learning based approach to to convert the base score of the Common Vulnerability Scoring System (CVSS) from  version 2.0 to 3.x.Once all CVSS 3.x scores are calculated, all vulnerabilities can be prioritised according to CVSS 3.x standard.

The authors need to describe the scoring mechanism in detail.

Reviewer 2 Report

This article deals with a very important topic. The study is devoted to the integration of various formats of databases of vulnerabilities. With ethos, the authors consider the CVSS database, although in fact the problem is much broader. In the future, you can use the research results to integrate not just different versions, but also different databases. The authors use machine learning methods and describe the process well. Nevertheless, I would like to expand the analysis of papers devoted to natural language processing and consideration of neurolinguistic methods.

Reviewer 3 Report

1. The paper does not fill a gap since this point motivating the work has not been sufficiently demonstrated due to the lack of sound statistical data and supportive references. The paper does not contain sufficient novelty to warrant publication. 

2. Regarding the background of this study, the discussion is shallow in the abstract. In fact, how to realize the novel algorithm should not be discussed in detail in this part. Change this focus in the abstract.

3. I think the mathematical analysis of the performance evaluation of the scheme is not enough, it should be futher strengthen.

4. The presentation should be improved and many typos and grammar errors should be corrected, such as missing an article before the word, the singular and plural forms of some words. You should check out the manuscript carefully. Besides, the variables need to be represented in italics.

5. The standard of references is not unified and the information of some references are incorrect.

more comments:

1. In the Introduction, you should point out the shortcomings of the previous research about CVSS 2.0 more specifically, in contrast to the method proposed by this paper.    2. You should add some structure details in Figure 1 to make the mechanism clearer to the reader. The current version is confusing and I have difficulty in understanding it. A more detailed description of Figure 1 should be given to clearly demonstrate the effectiveness of the obtained results.    3. The authors conducted their experiments with different algorithms, i.e. KNN, PNN, LIN, NB, and GRBF. What is the reason to choose these special algorithms? Necessary explanations for the selections are absent.     4. The article lacks description of algorithm for the aproposed method.  You should provide some details. And some parameters need to be clearly defined before used in those algorithms.

Round 2

Reviewer 3 Report

The revised paper has already met the quality requirements of the journal and is recommended to be accepted.